# A Mini Review of Citrus Rootstocks and Their Role in High-Density Orchards

**DOI:** 10.3390/plants11212876

**Published:** 2022-10-27

**Authors:** Faisal Hayat, Juan Li, Shahid Iqbal, Yang Peng, Leming Hong, Rashad Mukhtar Balal, Muhammad Nawaz Khan, Muhammad Azher Nawaz, Ummara Khan, Muhammad Asad Farhan, Caiqing Li, Wenpei Song, Panfeng Tu, Jiezhong Chen

**Affiliations:** 1College of Horticulture, Zhongkai University of Agriculture and Engineering, Guangzhou 510408, China; 2College of Forest Resources and Environmental Science, Michigan Technological University, Houghton, MI 49931, USA; 3Department of Horticulture, College of Agriculture, University of Sargodha, Sargodha 40100, Pakistan; 4Citrus Research Institute, Sargodha 40100, Pakistan; 5Key Laboratory of Food Processing and Quality Control, College of Food Science and Technology, Nanjing Agricultural University, Nanjing 210095, China; 6Institute of Applied Biology, Shanxi University, Taiyuan 030006, China; 7College of Horticulture, South China Agricultural University, Guangzhou 510642, China

**Keywords:** fruit trees, vigor, high density, dwarfing mechanism, hormonal signaling

## Abstract

Dwarfing is an important agricultural trait for intensive cultivation and effective orchard management in modern fruit orchards. Commercial citrus production relies on grafting with rootstocks that reduce tree vigor to control plant height. Citrus growers all over the world have been attracted to dwarfing trees because of their potential for higher planting density, increased productivity, easy harvest, pruning, and efficient spraying. Dwarfing rootstocks can be used to achieve high density. As a result, the use and development of dwarfing rootstocks are important. Breeding programs in several countries have led to the production of citrus dwarf rootstocks. For example, the dwarfing rootstocks ‘Flying Dragon’, ‘FA 517’, ‘HTR-051’, ‘US-897’, and ‘Red tangerine’ cultivated in various regions allow the design of dense orchards. Additionally, dwarf or short-stature trees were obtained using interstocks, citrus dwarfing viroid (CDVd) and various chemical applications. This review summarizes what is known about dwarf citrus rootstocks and the mechanisms underlying rootstock–scion interactions. Despite advances in recent decades, many questions regarding rootstock-induced scion development remain unanswered. Citrus rootstocks with dwarfing potential have been investigated regarding physiological aspects, hormonal communication, mineral uptake capacity, and horticultural performance. This study lays the foundation for future research into the genetic and molecular mechanisms underlying citrus dwarfing.

## 1. Introduction

Citrus fruits are one of the most popular tree fruits and are widely grown in tropical and subtropical regions around the world on a commercial scale [1,2]. Citrus fruits belong to the family Rutaceae, which consists of 140 different genera and 1300 different species, including oranges, mandarins, lemons, limes, pummelos, grapefruits, and several others [3]. Citrus fruits are known for their nutritional value, quality, aroma, and attractive flavor. Furthermore, they are an important source of vitamin C, dietary fibers, carbohydrates, and minerals [4,5,6,7,8].

Citrus has been commercially grown for hundreds of years using grafted plants, and rootstocks play a vital role in the growth and development of citrus plants [1]. Rootstocks influence the physiological and biochemical traits of scion cultivars, including plant vigor, quality, fruit production and tolerance against various environmental stresses [9,10,11]. It is believed that scions and rootstock are vital components in the fruit production industry, and their interplay regulates the supply of mineral nutrients, hormones, and carbohydrates [12]. Selecting a suitable rootstock is one of the major decisions in establishing an orchard and achieving maximum returns on a sustainable basis.

High-density (HD) planting systems are an innovative approach that helps improve yield, and net returns, particularly in the early stages of orchard development, by accommodating more plants per unit area than traditional planting systems [13]. Precocity, low cost per unit production and the potential for higher mechanization with improved input use efficiency are the main benefits of intensive cultivation systems. Since expanding the production space is limited, increasing productivity would help to facilitate production. Increasing production per unit area through agronomic management, i.e., high density is one of the efficient techniques to enhance the production of fruit crops [14,15]. In that regard, dwarfing trees have attracted the interest of citrus growers in different parts of the world due to their potential for higher planting density, easy harvest, pruning, and effective spraying, leading to improved productivity [16,17,18]. Flying Dragon (FD) is a size-controlling rootstock mutant of trifoliate orange (*Poncirus trifoliata* L. Raft var. monstruosa), which significantly reduces plant size when used as rootstock. Hence, most of the citrus scions with FD rootstock do not grow taller than 2.5 m, and overall growth may be reduced by 75% compared with trifoliate orange standard rootstock [19]. The high-vigor citrus rootstocks, such as ‘Rough lemon’ and ‘Volkamer lemon’, have been employed for commercial citrus production in several countries for many years. ‘Xiangcheng orange’, ‘Citrange’, and ‘Red tangerine’ rootstocks are commonly used in China and considered size-controlling (semi-dwarfing and dwarfing) [1]. Higher tree densities during the huanglongbing epidemic were also linked to lower disease incidence and more significant economic viability [20]. Moreover, a suitable planting density with restricted space available for the growth of these plants and avoidance of excessive intercrossing of scions must be employed [21]. In Brazil, high-density citrus orchards have 600–1250 plants ha^−1^, with 4–6 m distance between the rows and 2–3 m between the plants [22]. Long-term experiments in Japan have been conducted in the ‘Wase’ satsuma mandarin to assess orchards with higher densities of up to 10,000 plants ha^−1^ [23,24].

Taken together, this evidence suggests that high-density plantations are particularly important because this help improve the amount of fruit-bearing volume per hectare. The use of size-controlling rootstocks seems to be the primary option that enables the development of modern orchards under high-density planting systems. This paper aims to address the planting of high-density citrus trees using dwarfing rootstocks. The causes of dwarfism and mechanisms mediated by citrus rootstocks are also discussed.

## 2. Dwarfing Citrus Rootstocks

Citrus growers worldwide are attracted to dwarfing citrus rootstocks because they are ideal for high-density plantations and are suitable for mechanized farming [25]. Dwarfing citrus rootstocks are well represented in research reports (Table 1). Higher plant densities promote greater productivity; generally, lower densities permit the harvest of larger fruits, which raises the price of fresh fruit on the market [14]. Dwarf trees have several advantages, such as a better yield, high density, and photosynthetic efficiency, which raises potential production. In this system, plants will be trained in the assigned space, facilitating numerous practices such as harvesting, scouting, and spraying [26]. Additionally, high tree densities, in combination with adapted varieties, enable high-efficiency production techniques in many fruits [10,27,28,29].

It has long been believed that the ‘Flying Dragon’ trifoliate orange is the only true dwarfing rootstock in the citrus industry. Its commercial feasibility in tropical conditions has been established, particularly for more vigorous scion cultivars such as Persian lime and lemons [21,30]. Mature ‘Flying Dragon’ trees are typically about 2.5 m tall in most scion varieties. Conversely, this tree grows slowly when grafted to navel oranges, requiring several years to produce a commercial harvest. Hence, employing a dwarfing rootstock that grows faster and produces more fruit than scions grafted to ‘Flying Dragon’ is needed. However, the extensive use of Flying Dragon with sweet orange scion has not acquired commercial importance in the major producing areas, where farmers generally prefer more vigorous rootstocks [31]. As a result, most citrus breeding programs have developed new, alternative dwarfing rootstocks, and conventional cross-breeding has produced some promising genotypes [32,33] and genetic transformation [34].

### 2.1. Dwarfing by Chemical Treatments

Plant growth inhibitors are substances that slow down plant growth without altering developmental stages [42]. Many species are regularly treated with chemicals to control their height [43]. Plant growth regulators (PGRs), such as gibberellic acid (GA) biosynthesis inhibitors, are often used to limit excessive vegetative growth in various fruit crops, including apples, cashews, pomegranates, and citrus [44,45,46]. In the 19th century, Aron treated ‘Minneola’ tangelos (*Citrus paradisi* Macf.) with 1 g·L^−1^ of paclobutrazol before summer growth; the average shoot length decreased by nearly 50%. According to Garner et al. [47], prohexadione-calcium (P-Ca; 250 mg L^−1^) reduced the shoot growth of different fruit plants. Therefore, the PGRs approach should be further evaluated.

### 2.2. Dwarfing by Citrus Dwarfing Viroid (CDVd)

Plant dwarfism has been linked with several viruses and viroids [25,48,49]. The citrus exocortis viroid seems to cause dwarfism in citrus plants by increasing the aboveground hydraulic resistance [50]. Additionally, the rootstock, variety, and species of citrus hosts all affect the symptoms brought on by CDVd. CDVd infection of navel orange trees grafted on ‘trifoliate’ orange rootstock has revealed that the stunting phenotype caused by CDVd infection decreases canopy volume by about 50% [51,52,53,54]. Field research shows that the CDVd infection approach might be utilized to minimize plant height and maximize plantation density. Substantial findings were made on the possible biological mechanism by which some rootstock scion pairings are affected by CDVd to decrease tree canopy volume. Further information was revealed regarding the putative biological mechanism through which CDVd affects specific scion/rootstock combinations to reduce plant height [55]. According to Lavagi [25], CDVd has been used to produce dwarf citrus trees when grafted on trifoliate rootstock. Further findings demonstrate that CDVd modulates the expression profile of citrus growth and developmental processes, which may be responsible for reduced vegetative growth. However, the molecular mechanisms that restrict the canopy volume of citrus trees in response to CDVd infection are poorly understood. Hence, it seems that latent viruses could potentially contribute to regulating the dwarfing capacity of some rootstocks (Table 2).

### 2.3. Dwarfing by Using Interstocks

Interstock grafting is utilized in many fruit trees (Table 3), including citrus trees, as a sustainable approach to controlling plant height, dwarfing traits, and fruit quality [57,58,59]. According to previous studies, interstock and rootstock could be utilized jointly to overcome compatibility issues between the scion cultivar and rootstock [60]. When the ‘Flying Dragon’ rootstock is used as an interstock, it causes a considerable reduction of scion growth with both ‘Troyer’ citrange and ‘*P. trifoliata’* as rootstocks. Furthermore, compared to plants without interstock, the average size reduction is approximately 37%. However, using ‘Flying Dragon’ as a rootstock resulted in a 66% reduction in canopy growth compared with *P. trifoliata* and ‘Troyer citrange’ rootstocks [61].

Moreover, similar findings have been documented that lemon trees grafted with interstocks have smaller size, peel, and albedo thicknesses. Furthermore, interstocks affect the growth morphology and photosynthetic characteristics of ‘Yuanxiaochun’ grafted plants. In addition, when Kumquat and ‘Ponkan’ mandarin were employed as interstocks, the ‘Yuanxiaochun’ scion cultivar displayed greater photosynthetic activities and higher rates of light and CO_2_ utilization [62]. Interstocks influence the transport of water, nutrient uptake capacity, hormonal communication, and some other factors, and these interstocks influence overall plant growth, blooming, and fruiting. In addition, methods such as strangling, inarching, girdling, and grafting by budding are frequently employed throughout the interstocked-seedling production stages. Through stomatal and non-stomatal effects (girdling), these techniques can restrict photosynthetic carbon uptake and reduce transpiration [63,64,65].

### 2.4. Dwarfing by Using Tetraploid Rootstocks

In citrus, tetraploid trees can be used for the diversification of rootstocks because they have more genetic variability because of new recombination possibilities and their capability to serve as dwarf rootstock [70]. Tetraploids (4×), which result from incomplete mitosis of somatic embryos, might occur naturally or artificially in seedlings with diploid (2×) apomictic genotypes. Tetraploid rootstocks are characterized by shorter and thicker roots, which results in slower growth [71,72]. Furthermore, tetraploidy affects phenotypic features such as leaf and root morphology, fruit quality, stomatal size, and density. These alterations may disrupt normal physiological processes [73]. Tetraploid trifoliate orange rootstocks lowered scion canopy development and fruit yield; however, clementine’s sugar content, acidity, juiciness, and carotenoid content remained unaffected; hesperidin concentration increased, and this was only true for clementine scions grafted onto tetraploid rootstocks [74]. Allario et al. [75] evaluated diploid and tetraploid plants derived from the same seed (‘Rangpur’ lime; *C. limonia* Osbeck), and found that polyploid seedlings were smaller than diploid plants. According to Syvertsen [76], the lowest growth rates reported in citrus seedlings obtained from tetraploid rootstocks are attributed to decreased transpiration rates due to a lower number of stomata. Variation concerning plant height was noticed, and the diploid plants presented higher growth than tetraploid plants. Moreover, tetraploid plants were smaller and grew more slowly [72].

## 3. Dwarfing Mechanism of Scion Reduction

Grafting is an ancient horticultural practice that joins the aerial part (scion) with another segment (rootstock) to produce a new plant [10]. Scion cultivars grafted with rootstock are the foundation of modern fruit orchards. Rootstocks influence the morphological, biochemical, and physiological characteristics of the scion portion [77]. Several studies have been conducted to investigate the rootstock-induced dwarfing effect; however, the associated mechanisms in citrus plants have not been fully explained. Scion vigor is known to be influenced by multiple factors, such as the transport of minerals [2], level of hormones [78], hydraulic conductance [79], and anatomical studies [48,80]. Therefore, it can be concluded from the literature that the impact of citrus rootstocks on scion growth and dwarfing mechanisms are mediated by numerous factors (Figure 1 and Table 4), including mineral uptake capacity, hormonal alterations, hydraulic conductance, and anatomical features.

## 4. Type of Dwarf Rootstock

Dwarfing rootstocks produce a mature tree with a height of no more than 2.5 m, in combination with any scion cultivar, regardless of environmental influences [87]. The vigor of citrus trees (Citrus spp.) is affected by the canopy/rootstock combination, soil, and phytosanitary conditions. Bitters [19] proposed a classification in which a tree taller than 6.0 m was used as the standard. Sub-standard, semi-dwarf, and dwarf plants had a reduction of 25%, 50%, and 75%, respectively, regarding the standard. Another classification was proposed by Castle and Phillips [87] based on plant height or volume of scions into four different categories, such as standard plants: dwarf, semi-dwarf, semi-standard, and standard plants. Dwarf and semi-dwarf plants are 40% and 40–60% of the standard size, respectively. Semi-standard plants have 60–80% of the size of standard plants. On the other side, the term standard refers to plants having 80–100% of the standard size.

### 4.1. Tree Size and Vigor

Rootstocks significantly impact the physiological, biochemical, and molecular characteristics of the scion cultivar [16]. The reduction of scion growth due to rootstock is a fascinating phenomenon in studying fruit trees. Previous studies have demonstrated that the Salustiana scion cultivar grafted on ‘Rough lemon’ rootstock had the most extended primary shoot length, greater scion trunk diameter, and vigorous root morphology compared with less vigorous rootstocks. Additionally, plants grafted onto vigorous rootstocks have better nutritional properties [2]. The ‘Shatangju’ mandarin scion cultivar grafted onto the ‘Fragrant orange’ and ‘Red tangerine’ rootstocks displayed dwarfing traits with the shortest shoot length, lowest trunk diameter, and shortest internodal length [40]. In another study, the root system of ‘Rough lemon’ rootstock was shown to be vigorous with increased root projected area, root volume, surface area, and the number of forks and points; however, the ‘Carrizo’ rootstock displayed lower values of root morphological traits [81]. Recent experiments reported that the ‘Shatangju’ scion cultivar grafted onto the ‘Flying Dragon’ rootstock encouraged short-stature trees. In contrast, trees grafted with other rootstocks, such as ‘Shatang mandarin’, ‘Goutou sour orange’ and ‘Sour orange’, grew taller and wider and had more vigorous plant growth. According to the research mentioned above, the vegetative growth of scion cultivars is significantly influenced by citrus rootstocks. Additionally, using dwarfing rootstocks permits high-density planting, which boosts yield and leads to optimal use of water and nutrients [1].

### 4.2. Precocity

Prominent features imparted by dwarfing citrus rootstocks are a decrease in tree size and precocity (early flowering and fruiting). Dwarfing rootstocks are typically connected with precocity, while vigorous rootstocks delay fruiting. Conversely, the performance of the fruit trees is linked to a proper balance between fruiting and vegetative growth because excessive vegetative growth lowers the total yield and fruiting [88]. Rootstocks that encourage scion precocity are needed for early crop production [9]. For instance, dwarfing citrus rootstocks limit tree size and increase yield production and precocity. ‘Mandared’ trees grafted onto C22, C57, and C35 rootstocks bear fruit one year earlier than other tested rootstocks. Furthermore, ‘Mandared’ trees grafted with C22 rootstock demonstrated yield precocity and higher yield efficiency than C22 rootstock [89]. A lowered canopy volume of trees grafted on C22 rootstocks has been shown in previous studies [90,91], and could be an advantage for new plantings with higher densities.

## 5. Planting Density for Citrus Rootstocks

A high-density planting system (Table 5) is an innovative agrotechnology that enhances yield by managing more plants in a given area [92]. In addition, the appropriate plant density should be maintained for maximum yield and good-quality fruit [14]. Citrus trees in a grove compete for resources such as water, nutrients, and light. As the distance between trees decreases and resources become more limiting, competition increases, and there are notable tree responses [87]. A distance of 5–7 ft (1.52–2.13 m) is recommended between plants grafted onto Flying Dragon rootstock despite its limited commercial use in Florida [93]. In Southeast Brazil, 4–5 m row spacing and 1.5–2.5 m plant spacing are advised for the Flying Dragon rootstock [94]. In Japan, orchards of Wase satsuma mandarin with a density of up to 10,000 plants ha^−1^ were evaluated via long-term tests [23]. Recent research conducted in India with Nagpur mandarin on Rangpur lime rootstock determined that a high-density planting was regarded as one that included between 555 and 625 plants ha^−1^ and that an ultra-high-density planting was considered as one that contained between 1250 and 2500 plants ha^−1^ [13]. Therefore, long-term experiments will be needed to examine commercial citrus cultivars with dwarfing rootstocks to determine optimal plant density under modern production circumstances.

## 6. Conclusions

Grafting has been employed as a key tool in the propagation of horticultural crops to manage plant specific features such as early fruit production and vigor control. To encourage cultural application in high-density planting, citrus germplasm should be evaluated for its potential to produce plants with dwarf characteristics or short stature. In several countries, breeding programs have been started to develop dwarfing citrus rootstocks to achieve maximum planting density per unit area. For instance, ‘Flying Dragon’, ‘FA 517’, ‘HTR-051’, ‘US-897’, and ‘Red Tangerine’ rootstocks produce dwarf statures that allow for the establishment of dense orchards. Furthermore, short-stature trees have been obtained by using different interstocks and citrus dwarfing viroid (CDVd). Mechanisms for rootstock-induced dwarfing effects have also been covered in this article. Moreover, dwarf rootstocks have smaller root systems; thus, the roots absorb less water and nutrients from the soil medium. Furthermore, the influence of citrus rootstocks on scion growth and the dwarfing mechanism is induced by numerous factors, including mineral uptake capacity, hormonal alterations, hydraulic conductance, and anatomical features.

## Figures and Tables

**Figure 1 plants-11-02876-f001:**
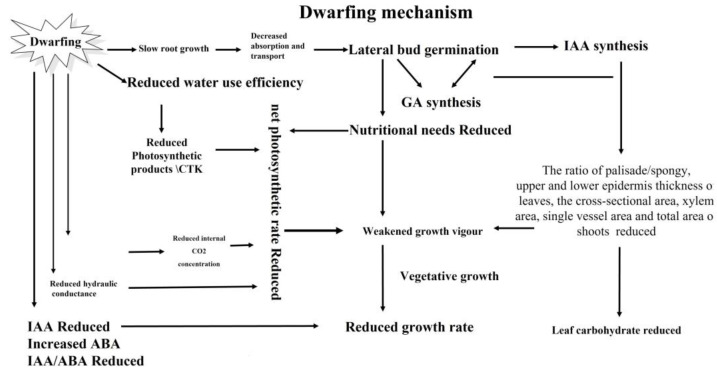
Schematic diagram of dwarfing mechanism in citrus rootstocks.

**Table 1 plants-11-02876-t001:** Main dwarfing citrus rootstocks are used worldwide.

Name of Rootstock	Origin	Parents	References
‘Flying Dragon’	Japan	*Poncirus trifoliata* var. monstrosa (T. Itô) Swingle	[35]
Forner-Alcaide (‘FA 418’)	Spain	Citrange ‘Troyer’ × *Citrus deliciosa* Ten	[32,36]
‘HTR-051’	Brazil	*Poncirus trifoliata* L. Raf. × *C. limonia*	[37]
‘US-897’	USA	Cross between Cleopatra mandarin × Flying Dragon	[38]
Forner-Alcaide (‘FA 517’)	Spain	King’ mandarin × *Poncirus trifoliata* (L.) Raf.	[32]
‘Ziyang Xiangcheng’	China	*Citrus junos Sieb. Ex Tanaka*	[39]
‘Red Tangerine’	China	*Citrus**reticulata* Blanco	[40]
‘Trifoliate Orange’	China	*Poncirus trifoliata*	[41]

**Table 2 plants-11-02876-t002:** Reduction of scion growth due to CDVd infection.

Scion Cultivar	Rootstock	Name of Treatment	Key Findings	References
‘Navel orange’	‘Rich 16-6’ trifoliate orange	Graft-inoculated with CDVd, while the control group was not inoculated with CDVd.	The tree canopy was reduced by >20% in CDVd-infected trees.	[51]
‘Parent Washington’ navel orange	‘Rich 16-6’ trifoliate orange	The CDVd-infected trees were planted at a close spacing (3 × 6.7 m), whereas the uninfected trees were planted at a standard spacing (6.1 × 6.7 m).	CDVd modifies the expression profile of citrus growth and developmental processes, which may be related to cellular changes that result in the observed phenotype of reduced vegetative growth and smaller trees.	[25]
‘Washington navel’	‘Carrizo citrange’	A graft was infected with viroid isolates.	Viroid infection had a negative impact on plant growth, resulting in decreased height and canopy volume.	[49]
‘Navel orange’	‘*Poncirus trifoliata*’	Trees were treated using citrus dwarfing viroid (TsnRNA-IIIb).	TsnRNAs (CDVd) can limit tree growth, making citrus grove management and production more flexible and consumer-friendly.	[56]
‘Grapefruit’	‘Troyer citrange’	The graft was inoculated with five different kinds of GTDC, together with 225T and 225M.	CVd infection of grafted grapefruit trees decreased the water movement capacity from the roots and within the canopy.	[50]
‘Valencia orange’	*‘Poncirus trifoliata’*	Treatment was performed using citrus viroid (CVd-la, CVd-IIIb and CVd-IIa,).	The canopy volume was reduced while the yield per tree increased.	[52]

**Table 3 plants-11-02876-t003:** Reduction of scion growth due to the interstock.

Scion Cultivar	Interstock	Rootstock	Key Findings	References
‘Yuanxiaochun’	‘Ponkan Shiranuhi’, ‘Hammi Taroceo’and ‘Kumquat’	‘Trifoliate orange’	The growth, development, and photosynthetic features of ‘Yuanxiaochun’ trees are significantly affected by interstocks. The grafted plant with ‘Shiranuhi’ as the interstock had the lowest values of morphological traits.	[62]
‘Navel orange’	‘Volkamer lemon’ and ‘Sour orange’	‘Volkamer lemon’ and ‘Sour orange’	Plants grafted on ‘Volkamer lemon’ had the tallest scion, longest roots, and most leaf numbers. Further, the scion stem recorded the highest contents of Mg, N, K, Fe, Zn, P, Mn, and phenols.	[66]
‘Mexican lime’	Different citrus types used as interstocks	‘Alemow’ (Mac)	‘Hiryu’ and ‘Flying Dragon’ rootstocks performed well when used as dwarfing interstocks. Moreover, the use of dwarfing trees allowed for the establishment of higher planting densities, reaching up to 600 trees per hectare.	[67]
*‘C. reticulata’*	*‘C. grandis’*	*‘C. hystrix’*	Compared to other combinations, citrus (Scion/rootstock/interstock/) combinations [‘*C. reticulata’*/’*C. aurantifolia’*/’*C. aurantium’*] produced lower values for morphological traits, i.e., plant height, root dry matter, etc. When ‘*C. grandis’* was used as the rootstock and ‘*C. hystrix’* as the interstock, there were no symptoms of HLB after six months of inoculation.	[60]
‘Satsuma mandarin’	‘Flying Dragon’ trifoliate orange	‘Flying Dragon’ trifoliate orange	High fruit soluble solids and size reduction are results of reduced sap flow in the scion cultivar caused by a heavy crop load and/or ‘Flying Dragon’ rootstocks or interstocks.	[68]
‘Salustiana orange’ (SAO), ‘Valencia Late’ (VLO)	‘Valencia Late’ grafted on CM/SO (CM/SAO/VLO)	‘Cleopatra mandarin’ (CM)	Interstock graft combinations had greater root growth than shoot growth, resulting in a lower shoot/root ratio than other combinations.	[69]

**Table 4 plants-11-02876-t004:** Rootstock-induced dwarfing effects in citrus scion growth.

Scion Cultivar	Rootstock	Traits	Key Findings	References
‘Shatangjyu’ mandarin	Eleven different types of rootstocks.	Hormone levels, scion growth, enzyme activity, and metabolite profile measurements.	‘Shantangju’ mandarin scion cultivar grafted onto ‘Flying Dragon’ rootstock produces dwarf plants. For high-density citrus cultivations, ‘Flying Dragon’ rootstock can be the best option under net house conditions.	[1]
‘Valencia’ sweet orange	Fifty-one different hybrid rootstocks.	Evaluations were made on tree strength and survival rate, fruit output and quality, drought tolerance, and graft compatibility.	Compared to conventional rootstocks, the scion canopy volume of three selected citrandarins of ‘Sunki’ mandarin (*C. sunki* (Hayata) hort. ex Tanaka) ‘Flying Dragon’ decreased by 70%.	[17]
‘Salustiana’ orange	‘Troyer citrange’, ‘Trifoliate orange’, ‘Rangpur lime’, ‘Carrizo citrange’, ‘Rough lemon’	Mineral analysis, scion vigor, and photosynthetic processes.	The primary shoot length of the ‘Salustiana’ scion grafted onto the Rough lemon rootstock was the longest. In addition, the ‘Rough lemon’ rootstock had a vigorous root system, which improved its ability to absorb minerals and nutrients.	[2]
‘Kinnow’ mandarin	‘Karna Khatta’, ‘Rough lemon’, ‘Rangpur lime’, ‘Troyer carrizo’, ‘citrange’, ‘sour orange’, ‘Jatti Khatti’	Mineral absorption capability and root morphology.	Higher mineral uptake was associated with a stronger root system (total root length, number of forks, and root tips), which can directly impact nutrient uptake.	[81]
‘Shatangju’ mandarin	‘Rough lemon’, ‘Fragrant orange’ ‘Canton lemon’, ‘Shatangju mandarin’, ‘Red tangerine’	Measurement of scion growth, RNA Seq, hormonal levels, qRT-PCR.	Scion vigor was significantly and positively associated with IAA and GA contents. Moreover, qRT-PCR revealed that IAA content and growth vigor were inversely linked with the expression levels of *ARF1*, *GH3*, *ARF8,* and *IAA4*.	[40]
‘Lemon’	‘Jatti Khatti’, ‘Attani-2’, ‘Rough lemon’, ‘RLC-4’, ‘Billikhichlli’, ‘Sour orange’, ‘Troyer citrange’ ‘Karna Khatta’,	Tee growth, yield, quality, and leaf nutrient concentrations.	Dwarfing rootstocks were less responsive to N, K, Ca, and Mg, and most of the micronutrients from the root medium.	[82]
‘Valencia’ orange	‘Flying Dragon’, ‘Rubidoux’	Production of biomass, hydraulic resistance, measures of gas exchange, xylem anatomy, and transport of 13C photoassimilates.	Rootstock-induced dwarfing in ‘Flying Dragon’ rootstock may be due to lowered hydraulic conductivity.	[79]
‘Lane Late’ navel orange	*‘C. macrophylla’*, ‘*C. volkameriana’* ‘Gou Tou Chen’, ‘Cleopatra mandarin’	Variables affecting fruit quality, yield, and growth.	Citrus rootstocks have a discernible impact on the fruit quality, yield, and vegetative growth of ‘Lane Late’ oranges. ‘Cleopatra’ mandarin and ‘Gou Tou Chen’ were the most suitable rootstocks for Lane Late scion cultivar under growing circumstances with thick and calcareous soil.	[83]
‘Kinnow’ mandarin	‘Rough lemon’, ‘Kinnow’, ‘Rangpur line’	Measurement of different morphological traits.	Among all evaluated rootstocks, ‘Rough lemon’ was found to be the most vigorous, while ‘Rangpur lime’ exhibited dwarf characteristics for Kinnow scion cultivar under agro-climatic conditions of Sargodha.	[84]
‘Marisol’ clementine	‘Swingle citrumelo’, ‘Sour orange, Carrizo citrange’, ‘Cleopatra mandarin/,	Measurements of vegetative growth parameters, yield, and fruit quality.	In high-density plantings under Egyptian circumstances, ‘Carrizo citrange’, ‘Swingle citrumelo’, and ‘Cleopatra mandarin’ are found to be effective rootstocks for ‘Marisol’ clementines.	[85]
‘Navelina;	Three different rootstocks (#23, #24, and F&A 418)	Vegetative growth, reproductive growth, hormonal analysis, photosynthetic measurements, and carbohydrate analysis.	The F&A 418 and #23 rootstocks result in dwarfing plants by promoting higher fruit and reproductive development, which slows summer vegetative growth.	[86]
‘Eureka’ lemon	‘Flying Dragon’, ‘Swingle citrumelo’, ‘Trifoliate orange’	Dry matter of different parts, endogenous IAA and ABA concentrations.	IAA levels were highest in the fresh shoots of ‘Swingle citrumelo’ and lowest in the ‘Flying Dragon’ rootstock.	[78]

**Table 5 plants-11-02876-t005:** Effects of planting density on plant growth and fruit quality.

Scion Cultivar	Rootstock	Planting Spacing/Density	Key Findings	References
Valencia	Kuharske citrange	358 and 955 trees/ha	High tree density, fertigation, and drip irrigation increased fruit yield in a CLas-infected sweet orange orchards.	[95]
Nagpur mandarin (*Citrus reticulata* Blanco)	Rangpur lime	Six different planting spacings.	Early and high yields were achieved through careful canopy management of plants planted with dense spacing (2 × 2 m, 2500 plant ha^−1^). From the first fruiting, the yield attained under the 2 × 2 spacing density was 26-times higher than the control.	[13]
Valencia sweet orange	IAC 1697, IAC 1710, Swingle, Swingle 4x	513, 696, and 1000 trees ha^−1^	Planting vigorous rootstocks at moderate to high tree densities improves the land-use efficiency of citrus orchards.	[31]
	Lime seedlings	Conventional density (400 plants ha^−1^), High density (800 plants ha^−1^), Ultra-High-Density (1600 plants ha^−1^)	High-density planting systems may be preferred during the early years of production.	[92]
Kinnow mandarin, Musambi	Lemon	Three planting distances: T1 (3.35 × 3.35 m), T2 (3.35 × 6.71 m) and T3 (6.71 × 6.71 m).	Plants planted at 11 × 22ft (T1) showed better results than other planting densities.	[96]
Kinnow mandarin	Rough lemon	6.00 m × 6.00 m, 6.00 m × 5.00 m, 6.00 m × 3.00 m	Vegetative growth of plants was greater at larger spacing (6.00 m × 6.00 m). Yield ha^−1^ was the maximum (220.99 t ha^−1^) at the closest spacing density (6.00 × 3.00 m).	[97]
Kinnow mandarin	Rough lemon	3.30 m × 6.60 m, 3.30 m × 3.30 m, 6.60 m × 6.60 m	Plant height and number of leaves were maximum at close planting density (3.30 m × 3.30 m). Moreover, the highest yield was observed at (3.30 m × 6.60 m) spacing density.	[98]

## Data Availability

Not applicable.

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
