# Peer review of "A Mini Review of Citrus Rootstocks and Their Role in High-Density Orchards"

_plants, 2022, doi:10.3390/plants11212876_

Round 1

Reviewer 1 Report

The article entitled "A comprehensive review on citrus rootstocks for high-density orchards" is a literature review of the different methods or factors to achieve citrus tree size reduction. This article is well documented with recent articles but needs to be complemented with the effect of polyploidization of rootstocks. There are several articles that deal with the reduction of tree size and volume by using tetraploid rootstocks (somatic hybrids or doubled diploids). 

Line 54 and 55: Rootstock varieties are named as being used for their dwarfing effect but no references are given. In a literature review this is not allowed and moreover these rootstocks are not known outside China. They are mentioned in Table 1 with a URL that does not open any web page. I propose to delete this sentence and remove these varieties from the table or put an article reference.

The citation of Table 1 in line 81 does not match with the sentence. It should be positioned elsewhere.

Line 91: it is not the rootstock that grows but the tree, i.e. the scion/rootstock association. Replace rootstock by tree.

In Table 1: ‘Flying dragon’ is a trifoliate orange tree or Poncirus trifoliata but certainly not of unknown parents. In the 7th row, a citrange is not of unknown parents it is a C. sinensis x P. trifoliata and in the last row the trifoliate orange is a full species P. trifoliata and not C. sinensis x P. trifoliata.

Line 104: what does PGRs mean?

Line 108: the g L-1 value is missing

Line 119: add orange to trifoliate (trifoliate orange)

In table 2: 'Rich 16-6' is a trifoliate orange it should be specified. Planting distances should be corrected in the second row (3.0 x 6.7 m) and (6.1 x 6.7 m)

Line 143 and 145: P. trifoliata in italics and line 145 clerical error (P. trifoliate)

There are many errors in the form of writing the names of varieties in the text and in the tables; sometimes there are quotation marks sometimes not. Sometimes they are not well placed. For example Troyer citrange should be written 'Troyer' citrange; Shatang Mandarin should be written 'Shatang' mandarin because group names are not capitalized like orange, lemon etc. When the name of a variety is mentioned, the name of the group should also be used; for example, line 199, Salustiana scion cultivar should be written 'Salustiana' sweet orange scion cultivar or line 207, the Carrizo rootstock should be written the 'Carrizo' citrange rootstock, etc.

Line 150: do not put quotation marks to Kumquat because it is not the name of a variety but the generic name of a species (Fortunella sp.) and to 'Ponkan' add mandarin

Table 3: Kumquat and not Kumqua. Flying Dragon' trifoliate orange and not trifoliate orange 'Flying Dragon'. In the rootstock column with Satsuma mandarin as scion, the interstock is 'Flying Dragon' but who is the rootstock? trifoliate orange? Correct the text. Define who is SO.

 Line 211: 'Goutou' is not an orange but a sour orange

Lines 220-222: This statement needs to be completed with a reference otherwise it is just speculation.

Line 230: 625 plants ha-1

Table 5: The international unit of distance is the meter and not the ft, so correspondences in m must be given.

The last sentence of the conclusion is out of context because it is a personal opinion of the authors on the expectations of CRISPR/Cas9 transformation and is not related to the theme of the article, dwarfism.

Two references (17 and 77) contain typing errors (capitalization instead of lower case).

Author Response

Responses to Reviewers

Title: A Comprehensive Review on Citrus Rootstock for High-Density OrchardsManuscript Number:  plants-1926755

Reply to Editors and Reviewers

We thank the referees for their interest in our work and comments that greatly helped us improve the manuscript, and we have tried our best to respond to the points raised. The Referees have brought up some good points, and we appreciate the opportunity to clarify our research objectives and results. We have now completely revised the manuscript. As indicated below, we have checked all the general and specific comments provided by the Referees and have made necessary changes accordingly.

Thank you very much for your attention and consideration.

Reviewer #1

(Note: Reviewer comments are in black text, and our responses are highlighted in blue color.

Comments

The article entitled "A comprehensive review on citrus rootstocks for high-density orchards" is a literature review of the different methods or factors to achieve citrus tree size reduction. This article is well documented with recent articles but needs to be complemented with the effect of polyploidization of rootstocks. There are several articles that deal with the reduction of tree size and volume by using tetraploid rootstocks (somatic hybrids or doubled diploids). 

Response: Thank you very much for your attention to reviewing this manuscript.

Line 54 and 55: Rootstock varieties are named as being used for their dwarfing effect but no references are given. In a literature review this is not allowed and moreover these rootstocks are not known outside China. They are mentioned in Table 1 with a URL that does not open any web page. I propose to delete this sentence and remove these varieties from the table or put an article reference.

Response: Thank you for your comment. We have added citations in the revised manuscript. (Please see line 80).

  1. The high-vigor citrus rootstocks, such as ‘Rough lemon’ and ‘Volkamer lemon’, have been employed for commercial citrus production in numerous countries for many years. Although ‘Xiangcheng orange’, ‘Citrange’, and ‘Red tangerine’ rootstocks are commonly used in China and are regarded as size-controlling rootstocks (semi-dwarfing and dwarfing) (Hayat et al., 2022).
  2. We have added references in mentioned places.

Table 1. Main dwarfing citrus rootstocks used worldwide.

Name of rootstock

Origin

Parentage

References

Flying Dragon

Japan

(Poncirus trifoliata var. monstrosa (T. Itô) Swingle

Cheng and Roose, 1995

Forner-Alcaide (FA 418)

Spain

Citrange ‘Troyer’ x Citrus deliciosa Ten

(Hervalejo et al., 2022; Martínez-Ballesta et al., 2010) [25,28]

HTR-051

Brazil

Poncirus trifoliata L. Raf. x C. limonia

(Marques et al., 2019) [29]

US-897

USA

Cross between Cleopatra mandarin × Flying Dragon

(Bowman et al., 2020) [30]

Forner-Alcaide (FA 517)

Spain

King’ mandarin x Poncirus trifoliata (L.) Raf.

(Hervalejo et al., 2022) [25]

Citrange

China

Citrus sinensis × Poncirus trifoliata

Song et al., 2020

Ziyang xiangcheng

China

Citrus junos Sieb. Ex Tanaka

Dong et al., 2019

Red Tangerine

China

Citrus reticulata Blanco

Liu et al., 2017

Trifoliate Orange

China

Poncirus trifoliata

Zhu et al., 2015

The citation of Table 1 in line 81 does not match with the sentence. It should be positioned elsewhere.

Response: Thank you for your comment. We have revised it. Please see lines 97-98.

Dwarfing citrus rootstocks are well represented in research reports (Table 1).

Line 91: it is not the rootstock that grows but the tree, i.e. the scion/rootstock association. Replace rootstock by tree.

Response: The suggested improvement has been made in a revised version (Please see line 156).

Conversely, when grafted to navel oranges, this tree grows slowly, requiring several years to produce a commercial harvest.

In Table 1: ‘Flying dragon’ is a trifoliate orange tree or Poncirus trifoliata but certainly not of unknown parents. In the 7th row, a citrange is not of unknown parents it is a C. sinensis x P. trifoliata and in the last row the trifoliate orange is a full species P. trifoliata and not C. sinensis x P. trifoliata.

Response: Thank you for your comment. Please see the revised draft (Table. 1).

According to your suggestion, we have corrected in 7th row (Citrus sinensis × Poncirus trifoliata) and in last row (Poncirus trifoliata).

‘Flying dragon’ is a trifoliate orange tree or Poncirus trifoliata but certainly not of unknown parents

Response: We have corrected it. Please see in the 1st row of table 1.

Line 104: what does PGRs mean?

Response: We have revised the text; we have added the complete name (Please see lines 169-172).

Many species are regularly treated with chemicals to control their height (Kulkarni, 1988). Plant growth regulators (PGRs), such as gibberellic acid (GA) biosynthesis inhibitors, are often used to limit excessive vegetative growth in various fruit crops, including apples, cashews, pomegranates, and citrus (Mog et al., 2019; Roux et al., 2010; Topp et al., 2012) [32–34].

Line 108: the g L-1 value is missing

Response: As suggested. We have added it (Please see line 174).

In the 19th century, Aron treated ‘Minneola’ tangelos (Citrus paradisi Macf.) with 1 g·L−1 of paclobutrazol before summer growth; the average shoot length was decreased by nearly 50%. According to Garner et al. (Garner et al., 2010).

Line 119: add orange to trifoliate (trifoliate orange)

Response: Added (Please see line 185).

In table 2: 'Rich 16-6' is a trifoliate orange it should be specified. Planting distances should be corrected in the second row (3.0 x 6.7 m) and (6.1 x 6.7 m)

Response: According to your suggestions, we have corrected it. Please see table 2.

The CDVd-infected trees were planted at a close spacing (3 × 6.7 m), whereas the uninfected trees were planted at a standard spacing (6.1 × 6.7 m).

Line 143 and 145: P. trifoliata in italics and line 145 clerical error (P. trifoliate)

Response: Corrected in both lines.

There are many errors in the form of writing the names of varieties in the text and in the tables; sometimes there are quotation marks sometimes not. Sometimes they are not well placed. For example Troyer citrange should be written 'Troyer' citrange; Shatang Mandarin should be written 'Shatang' mandarin because group names are not capitalized like orange, lemon etc. When the name of a variety is mentioned, the name of the group should also be used; for example, line 199, Salustiana scion cultivar should be written 'Salustiana' sweet orange scion cultivar or line 207, the Carrizo rootstock should be written the 'Carrizo' citrange rootstock, etc.

Response: According to your suggestions, we have revised our manuscript.

Line 150: do not put quotation marks to Kumquat because it is not the name of a variety but the generic name of a species (Fortunella sp.) and to 'Ponkan' add mandarin

Response: Corrected. Please see line 275.

Table 3: Kumquat and not Kumqua. Flying Dragon' trifoliate orange and not trifoliate orange 'Flying Dragon'. In the rootstock column with Satsuma mandarin as scion, the interstock is 'Flying Dragon' but who is the rootstock? trifoliate orange? Correct the text. Define who is SO.

 Response: We apologize for the mistake. We have corrected it.

The correct name is Kumquat, and in the 2nd line interstock and rootstock, both are ‘Flying Dragon’ trifoliate orange.

Line 211: 'Goutou' is not an orange but a sour orange

Response: Corrected (line 366).

Lines 220-222: This statement needs to be completed with a reference otherwise it is just speculation.

Response: We have revised this statement with reference. Please see lines 392-395.

Citrus trees in a grove compete for resources like water, nutrients, and light. As the distance between trees decreases and resources become more limiting, competition increases, and there are notable tree responses (Castle and Baldwin 1996).

Line 230: 625 plants ha-1

Response: Corrected. Line 402.

Table 5: The international unit of distance is the meter and not the ft, so correspondences in m must be given.

Response: Thanks for the correction. We have corrected it.

Three planting distances: T1 (3.35 x 3.35 m), T2 (3.35 x 6.71 m) and T3 (6.71 x 6.71 m).

The last sentence of the conclusion is out of context because it is a personal opinion of the authors on the expectations of CRISPR/Cas9 transformation and is not related to the theme of the article, dwarfism.

Response: We have deleted this sentence from the conclusion part.

Two references (17 and 77) contain typing errors (capitalization instead of lower case).

Response: Corrected references 17 and 77.

Reviewer 2 Report

See attached review.

Author Response

Responses to Reviewers

Title: A Comprehensive Review on Citrus Rootstock for High-Density OrchardsManuscript Number:  plants-1926755

Reply to Editors and Reviewers

We thank the referees for their interest in our work and comments that greatly helped us improve the manuscript, and we have tried our best to respond to the points raised. The Referees have brought up some good points, and we appreciate the opportunity to clarify our research objectives and results. We have now completely revised the manuscript. As indicated below, we have checked all the general and specific comments provided by the Referees and have made necessary changes accordingly.

Thank you very much for your attention and consideration.

Reviewer #2

Note: Reviewer comments are in black bold text, and our responses are highlighted in blue color.

This manuscript is a brave attempt at reviewing challenging subjects. There are many bits and pieces of citrus research on the subjects of planting density/spacing and appropriate rootstocks, but rarely when the literature is assembled do they constitute what most would agree is a comprehensive story. These subjects have not been thoroughly studied as noted by the authors. Therefore, it is noteworthy that they are occasionally reexamined and sometimes investigated again. It is for these reasons that I submit the following assessments that if accepted could improve the manuscript substantially.

Response: Thanks for reviewing our manuscript.

  1. The manuscript does not warrant the use of “Comprehensive” in the title. Perhaps “A Mini review of Citrus Rootstocks and their Role in High Density Orchards.” There are many significant publications from Australia, Florida and California that have been almost entirely excluded. Also, Agromillora Company in Barcelona, Spain is a strong advocate doing their own research on HD plantings with rootstocks developed in Spain.

Response: According to your suggestion, we have changed the title.

“A Mini review of Citrus Rootstocks and their Role in High-Density Orchards”

  1. An examination of published reports requires also addressing the question: Given all the research, has it been convincing? Answer: No. The authors need to make a strong statement about the fact that even though there is a moderate amount of research with positive outcomes as presented in their review, high density planting and dwarfing rootstocks are not especially important commercially worldwide with a few exceptions. Why?

Response: Dwarfing citrus rootstocks are very important for modern citrus orchards because they are ideal for high-density plantations and easy for mechanized farming (Lavagi- 76 Craddock et al., 2022) [17]. Higher plant densities promote greater productivity; generally, lower densities permit the harvest of more large fruits, which raises the price of fresh fruit on the market (Haque et al., 2022) [18].

  1. It is not clear why the review was drafted. Stronger, clear statement of objectives. What did the authors attempt to accomplish

Response: We have revised our objectives.

Furthermore, dwarf or short-stature trees were also obtained through the use of interstocks, citrus dwarfing viroid (CDVd), and the applications of different chemicals. This paper provides an overview of what is known about dwarf citrus rootstocks and the mechanisms underlying rootstock–scion interactions. Despite the improvements made during the past decades, numerous questions regarding rootstock-induced scion development remain unanswered. In addition, citrus rootstocks with dwarfing potential were approached, including physiological aspects, hormonal communication, mineral uptake capacity, and horticultural performance.

  1. An argument for HD plantings should appear early in the manuscript; in the introduction.

Response: As suggested (Please see lines 66-75).

  1. In several place [like the Intro], the flow of thoughts is jumpy, i.e., the statements do not transition smoothly from one point to the next.

Response: We have revised it (Please see lines 66-84).

  1. The list of rootstocks in Table 1 should meet certain criteria for dwarfing or size-controlling. A key one is consistency. Do we know that about Cunningham which is not really a commercial rootstock in California.

Response: We are sorry for this mistake; we have deleted Cunningham rootstock from table 1.

  1. There is an important, but subtle difference between dwarfing rootstock and size-controlling rootstock.

Response: We have made it uniform in the whole manuscript according to your suggestion.

  1. Precocity is a mechanism of dwarfing not discussed in the review. We have experimented with Rangpur x Troyer citrange rootstock. It is very precocious and fruits early.

Response: We have added this part in a revised version. (Please see lines 372-387).

Precocity

Prominent features imparted by dwarfing citrus rootstocks are a decrease in tree size and precocity (early flowering and fruiting). Dwarfing rootstocks are typically connected with precocity, while vigorous rootstocks delay the fruiting. Conversely, the performance of the fruit trees is linked with proper balance among fruiting and vegetative growth, because excessive vegetative growth lowers the total yield and fruiting (Atkinson and Else, 2001). Rootstocks that encourage scion precocity are needed for early crop production (Castle, 2010). For instance, dwarfing citrus rootstocks limit tree size and increases yield production and precocity. ‘Mandared’ trees grafted onto C22, C57, and C35 rootstocks started to bear fruit one year earlier compared with other tested rootstocks. Furthermore, ‘Mandared’ trees grafted with C22 rootstock demonstrated yield precocity and high yield efficiency than C22 rootstock (Caruso et al., 2020). Lowered canopy volume of trees grafted on C22 rootstock has been shown in previous studies (Continella et al., 2018; Siebert et al., 2010), and could be an advantage for new plantings with higher densities. Earlier studies has demonstrated that trees grafted onto C22 rootstock had a smaller canopy volume (Continella et al., 2018; Siebert et al., 2010), which may be valuable for new plantings with high densities.

  1. Dwarfing rootstocks tend to produce trees with smaller fruit. Is that important? Yes.

Response: Yes, in previous studies, it has been reported that dwarfing rootstocks produce smaller fruits compared with trees grafted on vigorous rootstocks.

Legua et al. (2011) established that rootstock significantly affected fruit quality variables. C. macrophylla and C. volkameriana would appear to encourage the highest fruit weight.

Akber Hayat et al. (2019) reported that plants budded with C-35 rootstock attained minimum plant height and lowest fruit weight compared with other tested rootstocks. Moreover, Feutrell’ early budded on different rootstock achieved maximum plant height on Cox mandarin rootstock, followed by Troyer citrange and Rough lemon rootstocks. This study support the above statement that dwarfing rootstocks tend to produce trees with smaller fruit. In another study, maximum fruit size of Kinnow was measured on Poncirus trifoliata, followed by Fraser hybrid, while fruit diameter of Kinnow was lowest on Troyer citrange rootstock. Similar results were observed in a research trial and significant effects of different rootstocks on fruit size of citrus were noted, Goutou sour orange produced more large sized fruits as compared to other rootstocks (Louzada et al., 2008). Jaskarni et al. (2002) discovered that diploid kinnow trees were much better than tetraploid as for as fruit weight was concerned.

  1. An approach the authors may wish to take is to briefly review the main elements on planting density and spacing and then write about what is new or has changed/improved in the last decade.

Response: Thank you for your suggestion. In the future, we will follow your instructions

Round 2

Reviewer 1 Report

There are several articles that deal with the reduction of tree size and volume by using tetraploid rootstocks (somatic hybrids or doubled diploids). This factor of dwarfing is missing in the  article the authors have to complete it

Author Response

We thank the Referees for their interest in our work and for helpful comments that will significantly improve the manuscript

 (Note: Reviewer comments are in black text, and our responses are highlighted in blue color.

Reviewer #1:

Comments

There are several articles that deal with the reduction of tree size and volume by using tetraploid rootstocks (somatic hybrids or doubled diploids). This factor of dwarfing is missing in the article the authors have to complete it

Response: In the revised version, we have added necessary literature related to tetraploid rootstocks (somatic hybrids or doubled diploids). Please see lines 183-203.

Reviewer 2 Report

See separate document with original review and review of revision.

Author Response

We thank the Referees for their interest in our work and for helpful comments that will significantly improve the manuscript

 (Note: Reviewer comments are in black text, and our responses are highlighted in blue color.

 Reviewer 2

A Mini-Review of Citrus Rootstocks and their Role in High-Density Orchards Review of revised manuscript 2 Oct 22 Bill Castle The authors made a good attempt at revising manuscript, but there are areas still needing improvement.

  1. The authors are to be applauded for their efforts to convey the information and subtleties of a language not probably native to most or all of them. I suggest a closer review of grammar and clarity. For example, these first two sentences in the Intro: “Citrus fruits are frequently grown for consumption around the world (Hayat et al., 46 2022; Khan et al., 2020) [1,2]. These are favored due to their superior quality, aroma, large fruit size, higher production, and greater adaptability to various agro-climatic conditions and soils (Shireen et al., 2018) [3].“ The first sentence should declare that citrus is grown in many places as a commercial crop exceeded in value only by banana [or whatever is correct today]. Then, citrus fruit are well known for their high quality, aroma and flavor. Large fruit size? Larger than what? Other commercial fruit crops? Same question for higher production and adaptability. Further, Horticultural crops have been grown via grafting regularly for hundreds of years (Hayat et al., 2021a) [4]. Fruits are complicated objects for investigating the shoot-root interaction since recent fruit farming systems primarily depend 51 on scion variety, which is grafted on rootstocks (Nawaz et al., 2016; Warschefsky et al., 52 2016) [5,6]. These two sentences seem to just be hanging there with no established connection to your story. There is no flow in building background. You need to tell a story that flows from citrus fruit have been around a long time and are important commercially, to production systems and cultivation and the continued/expanding interest in HD plantings. Why? See the Fla. State Hort. Society Proceedings for 1976 and a presentation by Herman J. Reitz on why. Then explain why rootstocks are critical. You cite a publication from Brazil in which the author[s] state that planting density/spacing is all about matching tree vigor and site conditions with spacing. That is exactly right. It’s what I call the Sweet Spot. Thus, size-controlling rootstocks [a term I prefer] are needed. That’s basically your story. Don’t make this more complicated than necessary.

Response: We have thoroughly revised the introduction section of this manuscript; please check the revised version (Lines 44-95).

  1. You have not defined a HD planting.

Response: We have added this. Please see lines 61-63.

High-density (HD) planting system is an innovative approach that helps improve yield and net returns, particularly early stages of orchard development, by accommodating more plants per unit area than a traditional planting system (Ladaniya et al., 2021).

  1. HD plantings and mechanical harvesting? That has to be part of the modern story.

Response: We have added the necessary information. Please see lines 61-69.

High-density (HD) planting system is an innovative approach that helps improve yield and net returns, particularly early stages of orchard development, by accommodating more number of plants per unit area compared with a traditional planting system (Ladaniya et al., 2021). Precocity, low cost per unit production and the potential for higher mechanization with improved input use efficiency are the main benefits of intensive cultivation systems. Since expanding the production space is limited, increasing productivity would help to facilitate production. Increasing production per unit area through agronomic management, i.e., high-density is one of the efficient techniques to enhance the production of fruit crops (Haque et al., 2022).

  1. Again, there is way too much important literature not cited. Studying that literature is important to have a proper background and perspective on the subject to write even a minreview. There is a strong argument for also introducing more apple literature and contrasting those reports with citrus orchards.

Response: We have revised this section. Please see the introduction part.

  1. I don’t think the interest in dwarfing [size-controlling] rootstocks is nearly as universal as you’ve stated. HD orchards have been and remain a matter mostly of local circumstances. An excellent example is Australia where some fine research was conducted to develop a production system of closely spaced trees inoculated with viroids. Despite the system 2 being thoroughly researched, it was not adopted commercially. It required a susceptible rootstock and growers were not willing to change.

Response: We have added necessary literature about high-density and discuss the role of dwarfing rootstock in high-density plantations (Please see lines 61-95).

  1. The emphasis in the manuscript seems to be on Chinese rootstocks and planting systems. If you wanted to change your story, a mini-review of Chinese citrus history in relation to rootstocks and HD would be quite interesting. That would provide an opportunity to introduce the plant materials and cultural background [growing techniques and social aspects].

Response: Thanks for your suggestions. We think only a few dwarfing rootstocks are available worldwide, so the general title will be more suitable to cover the entire contents.

  1. Table 1. Were citranges made in China? I think so naturally, but not those man-made. Also, there are many citranges and citrumelos. Not all are size-controlling. You should state which ones are size-controlling.

Response: We apologize for the mistake; we have removed the citrange rootstock part from table 1.

Round 3

Reviewer 1 Report

The manuscript is now publishable